# The Role of cMET in Gastric Cancer—A Review of the Literature

**DOI:** 10.3390/cancers15071976

**Published:** 2023-03-26

**Authors:** Filip Van Herpe, Eric Van Cutsem

**Affiliations:** Department of Digestive Oncology, University Hospitals Gasthuisberg, 3000 Leuven, Belgium

**Keywords:** cMET, gastro-oesophageal cancer, gastric cancer, review

## Abstract

**Simple Summary:**

cMET is a proto-oncogene that has been extensively studied in gastric cancer. Gastric cancer (GC) is a heterogenous disease with varied histology and molecular profiling. It still implies a poor prognosis in stage IV. New targeted therapeutic options are being investigated. In this review, we analyzed all studies performed on gastric cancer with MET-inhibitors. In first-line therapy, the addition of MET-inhibition to chemotherapy did not show any benefit in allcomers. Different tyrosine kinase inhibitors (TKI) have been investigated in small cohorts with different diagnostic assays added to the inclusion criteria. Determining patients with gastric cancer who benefit from cMET inhibitors remains difficult. Potentially only *MET* amplification detected by comprehensive genomic testing could be a good targeted option, although the prevalence is limited to less than 5% of all patients with gastric cancer.

**Abstract:**

Gastric cancer (GC) is an important cause of cancer worldwide with over one million new cases yearly. The vast majority of cases present in stage IV disease, and it still bears a poor prognosis. However, since 2010, progress has been made with the introduction of targeted therapies against HER2 and with checkpoint inhibitors (PDL1). More agents interfering with other targets (FGFR2B, CLDN18.2) are being investigated. cMET is a less frequent molecular target that has been studied for gastric cancer. It is a proto-oncogene that leads to activation of the MAPK pathway and the PI3K pathway, which is responsible for activating the MTOR pathway. The prevalence of cMET is strongly debated as different techniques are being used to detect MET-driven tumors. Because of the difference in diagnostic assays, selecting patients who benefit from cMET inhibitors is difficult. In this review, we discuss the pathway of cMET, its clinical significance and the different diagnostic assays that are currently used, such as immunohistochemy (IHC), fluorescence in situ hybridization (FISH), the H-score and next-generation sequencing (NGS). Next, we discuss all the current data on cMET inhibitors in gastric cancer. Since the data on cMET inhibitors are very heterogenous, it is difficult to provide a general consensus on the outcome, as inclusion criteria differ between trials. Diagnosing cMET-driven gastric tumors is difficult, and potentially the only accurate determination of cMET overexpression/amplification may be next-generation sequencing (NGS).

## 1. Introduction

Gastric cancer (GC) is an important cause of cancer worldwide with over one million new cases per year and, in 2020, an estimated 769,000 deaths [1].

Gastric adenocarcinoma is a heterogenous disease and is categorized in different subgroups based on histology (diffuse vs. intestinal), as well as on molecular profiling, microsatellite instability (MSI), Epstein–Barr virus (EBV), genomic stability (GS) and chromosomal instability (CIN). Apart from immunotherapy-sensitive MSI or EBV-driven gastric cancer, the prognosis in stage IV disease remains poor.

The first-line standard of care therapy remains the doublet chemotherapeutic combination of platinum-based therapy (cisplatin or oxaliplatin) in combination with a fluoropyrimidine (5-fluorouracil, capecitabine or S-1). Occasionally, a third cytotoxic agent is added (usually docetaxel). Over the last 15 years, molecular-driven targeted therapy has evolved rapidly. Patients with metastatic GC can be subdivided into two groups based on HER2 expression status: HER2 positive disease, meaning a 3+ score on protein immunohistochemistry (IHC) and a 2+ score on IHC followed by fluorescence in situ hybridization (FISH) of ≥2. For patients with HER2 positive disease, trastuzumab can be added to standard of care chemotherapy as the ToGA trial showed improved progression-free survival (PFS) and overall survival (OS) in HER2-positive patients [2].

In 2015, VEGFR-2-inhibition was added in the second line with ramucirumab for gastric cancer in combination with paclitaxel [3]. It showed an improved OS compared to placebo as well as paclitaxel monotherapy. Recently, in 2021, the addition of a checkpoint inhibitor (the PD-1 antibody nivolumab) was added in HER2 negative patients harboring a PDL1 CPS score > 5 on tumor samples in first-line gastric cancer in combination with platinum-based doublet chemotherapy [4]. A combined treatment of another checkpoint inhibitor, pembrolizumab, with trastuzumab, in HER2 positive patients, showed a very high response rate (ORR) in the recent findings of the Keynote 811 trial [5]. However, in PDL1 and HER2 negative populations, there is still a need for improvement. Due to increased comprehensive genomic analysis of tumor DNA and the development of new targets, more targeted therapies are being found in gastric cancer. This includes targets such as CLDN18.2 and FGFR2b overexpression, and the landscape is currently still expanding. One of the potential targets in gastric cancer is cMET. *MET* is a proto-oncogene which encodes for a transmembrane receptor with tyrosine kinase activity. In this review, we discuss how diagnostic testing of *MET* overexpression/amplification is performed in gastric cancer, and, next, provide an overview of all clinical studies that have been published with cMET inhibitors in gastric cancer.

## 2. Materials and Methods

A literature search was performed of all studies published from 2008 until July 2022 on Pubmed (https://pubmed.ncbi.nlm.nih.gov/, accessed on 21 November 2022) as well as on clinical trials.gov (https://clinicaltrials.gov/, accessed on 21 November 2022) including cMET in gastric cancer. Search terms such as “Gastric cancer”, “cMET”, “gastro-esophageal cancer” and “c-MET” were used in different combinations. All background information regarding cMET was found by selecting articles from the search term: “cMET in gastric cancer” on Pubmed. From the 322 articles found, 51 were selected to include in this review.

## 3. Results

### 3.1. General Background and Preclinical Data on cMET in Gastric Cancer

*MET*, also known as the N-methyl-NO-nitroso-guanidine human osteosarcoma transforming gene, was originally discovered in 1984 by Cooper et al. working on osteosarcoma [6]. The *MET-gene* is located on chromosome 7q21-31 and consists of a heterodimer with a small extracellular alpha-chain subunit (50 kDa) and a larger single-pass transmembrane beta-chain subunit (145 kDa). The extracellular domain contains three functional domains: the SEMA domain and the plexin-semaphoring-integrin (PSI) domain, together with four immunoglobulin-like regions in plexins and transcription factors (IPT 1–4). The intra-cellular part is subdivided into three domains: a juxtamembrane (JM) domain, a tyrosine kinase (TK) domain and a C-terminal multi-functional docking site (MFDS). These are all regulated by phosphorylation. Phosphorylation of the JM domain results in inhibition of the kinase domain and degradation of cMET, whereas phosphorylation of the TK domain results in upregulation of the kinase activity of cMET [7]. Phosphorylation in the MFDS domain directly mediates recruitment of downstream signaling molecules, such as SHIP2, PIR3K, GRB2, GAB1, etc. [8,9,10,11].

cMET can become activated through homodimerization upon binding of HGF or “hepatocyte growth factor” through its HGF-ligand binding sites (IPT3–4 and SEMA domains), which leads to phosphorylation and activation of downstream signaling. This process is also called the *canonical activation* of cMET. Alternatively, cMET can be activated in an HGF-independent manner, so-called *non-canonical activation*. For example, des-gamma-carboxyl prothrombin (DCP) is shown to induce cell proliferation via cMET-Janus kinase 1-STAT3 signaling by causing auto-phosphorylation in the cMET-PK domain in hepatocellular carcinoma [12]. cMET activity can also be monitored by the interaction of several *signal modifiers*. Integrin α6β4 potentiates HGF-triggered activation of RAS and PI3K [13]. Class B pexin transactivates cMET in response to stimulation of semaphoring and induces the execution of cMET-dependent biological responses [14]. Transmembrane cell adhesion molecules of the CD44 family link the cMET cytoplasmatic domain to actin microfilaments via growth factor receptor bound protein 2 (GRB2), also facilitating cMET induced activation of RAS via the son of sevenless protein (SOS). The FAS receptor (FAS-R) also interacts with the cMET extracellular domain, thereby preventing FAS-R and FAS ligand recognition, FAS self-aggregation, and limiting apoptosis through the extrinsic pathway [15]. Functional interactions have also been described with epidermal growth factor receptors (EGFR) enabling activation of cMET after cellular stimulation by EGFR. cMET can even be stimulated by EGFR in the absence of HGF and the simultaneous activation of cMET and EGFR is synergistic. cMET can, conversely, also upregulate EGFR ligands. There is also evidence that ERBB2 and ERBB3 receptors can cause transactivation through cMET, which could help induce resistance to targeted therapies and strengthen downstream pathways, such as Akt and ERK/MAP kinase [16,17,18,19,20]. Apart from receptor- or ligand-driven activation, cMET can also be activated by hypoxia, inactivation of tumor suppressor genes, microRNAs, autocrine cMET induced activation, etc. [21].

The first relation between cMET and gastric cancer (GC) was described in 1992 in eleven gastric cell lines that showed the presence of *MET* amplification on chromosome 7 in mostly diffuse type gastric cancer and was indicative of poor prognosis [22]. In a separate study, about 18% of 154 gastric cancers included stained positive for cMET and showed more prevalent MET expression in more advanced GC [23]. Other studies in gastric cancer cell lines showed that anti-HGF inhibited further cell growth and that HGF could be produced by gastric fibroblasts, the invasiveness of which could be promoted by MET expression on gastric cancer cells [24,25]. In 2011, Toiyama et al. investigated the co-expression of HGF and MET as a predictor for peritoneal dissemination. In 100 patients with gastric cancer, the expression of HGF and MET was higher in patients with more advanced disease as well as peritoneal metastasis showing that it plays an important role in epithelial-mesenchymal transition [26]. Comprehensive molecular characterization in gastric cancer revealed by mRNA sequencing alternative splicing events of the *MET 2* exon showed skipping in 30% of cases, which also resulted in MET overexpression. In 17% (47/212) of gastric cancers, new variants of MET were skipped in exon 18 and 19. The removed exons encoded regions of the kinase domain. In up to 8% of cases, *MET* amplification could be detected [27]. Peng et al. published a systematic review with meta-analysis that included 2258 patients with gastric cancer in a total of 16 studies that provided data on *MET* expression and amplification. Although these data were subject to a publication and selection bias, overexpression and/or amplification of MET was associated with poorer survival [28]. The combination of this data together led to the further development of cMET inhibitors in gastric cancer as new potential molecular drivers [29]. An overview of the cMET-pathway and its therapeutic landscape is given in Figure 1.

The cMET tyrosine receptor kinase can be activated through protein overexpression, gene amplification, increased HGF ligand autocrine expression, enhanced paracrine ligand-mediated stimulation, inadequate cMET degradation, ligand-independent activation and, rarely, gene mutation. In gastric cancer, *HGF/cMET mutations* are exceedingly rare. Activation of MET in gastric cancer is thought to be primarily a result of receptor overexpression and/or genomic upregulation (amplification/fusion) [30].

Over the years, different methods have been developed to detect MET overexpression and/or amplification in gastric cancer, all of which have their advantages/disadvantages. *MET* protein expression can be evaluated by immunohistochemistry (IHC), with confirmation of in situ hybridization (ISH), or by next generation sequencing (NGS). The most successful clinical experience comes from non-small-cell lung cancer (NSCLC) in which *cMET* alterations are found in 3–4% of all patients. The *MET* exon 14 skipping mutation is most common and, more rarely, also MET amplification can be found. In the past, protein immunohistochemistry (IHC) and mass spectrometry have been used to detect *MET* exon 14 mutations; however, it has been shown to be unreliable as a screening tool for *MET*-exon-14-positive patients. Sanger sequencing or RT-PCR can be used to detect *MET* alterations, and, in practice, next generation sequencing (NGS) is performed [31].

#### 3.1.1. Immunohistochemistry (IHC) and Fluorescence In Situ Hybridization (FISH)

Diagnostic determination of protein overexpression by immunohistochemistry (IHC) and fluorescence in situ hybridization (FISH) has a long history in HER2-positive breast and gastric cancer [32].

Several commercial MET antibodies exist to determine protein overexpression by IHC, although comparison of the performance has not yet been performed as of today. To detect MET protein expression by IHC, a slide with hematoxylin and eosin-stained sections is selected. The most commonly used antibody is SP44, a rabbit monoclonal anti-total MET antibody clone. The first test for scoring MET overexpression is performed by analysis of the percentage of positive tumor cells (scale 0–100%) with a staining intensity of 0 to 3+: negative (0), weak (1+), moderate (2+) or strong (3+). A cut point of >50% of tumor cells staining moderately or strongly has been associated with treatment benefit in NSCLC and is often used as a standardized cut-off of MET overexpression.

The second immunohistochemical technique for scoring MET overexpression is evaluated using the H-score. The H-score multiplies the percentage of cells with 1+, 2+ or 3+ staining by the percentage of positive cells (from 0% to 100%). The H-score ranges from 0–300 with ≥200 indicating overexpression, although the cut points do vary between studies [33,34,35,36]. However, MET IHC overexpression does not strongly correlate with *MET* amplification. This can be explained by the inclusion of lower levels of *MET* amplification not causing substantial protein expression, or expression that is being controlled post-transcriptionally [37].

*MET* gene amplification is commonly assessed with an in situ hybridization technique (ISH) by a MET/CEP7 dual color probe set. This technique was developed to distinguish polysomy from true amplification, as polysomy typically does not result in response to targeted therapy. In *MET* polysomy every additional chromosome 7 will have a corresponding MET-gene location and this will not result in increase in the MET/chromosome 8 centromere ratio (MET/CEP7). In *MET* amplification, there will be an increase in MET copies on chromosome 7, resulting in an increase in the MET/CEP7 ratio. *MET* gene amplification can be defined using FISH or by gene copy number (GCN) > 5 based on the Cappuzzo criteria [38]. Alternative criteria include a MET GCN of ≥6 and a MET GCN of ≥15, although when determining only gene copy numbers, these do not enable differentiation between polysomy and true focal amplification because other zones of the chromosome are not searched for, and the absolute number of MET-containing chromosomes cannot be determined. For that reason, a ratio between MET and CEP7 can, therefore, enable focal amplification. Depending on the literature, a MET/CEP7 ratio ≥ 2.0 or 2.2 is chosen. In a different study, a categorization of the degree of amplification was performed in three groups based on MET/CEP7 ratios: low ≥ 1.8 to ≤2.2; intermediate > 2.2 to <5; and high ≥ 5. In NSCLC, *MET* amplification (MET/CEP7 ratio > 2.2) was only detected in 1% of patients with MET overexpression (H score ≥ 200). In gastric cancer, however, a MET IHC H-score of 150 had a 75% sensitivity and 78% specificity to detect *MET* amplification (MET/CEP7 ratio > 2.0 and GCN > 4.0) showing the difficulty of establishing a correlation between protein overexpression and its ability to detect *MET* amplification [39].

#### 3.1.2. Next-Generation Sequencing (NGS)/Comprehensive Genomic Profiling (CGP)

No standardized copy number nor cutoff has been determined as of today for next-generation sequencing. *MET* amplification can be detected by next-generation sequencing (NGS) but strongly depends on the quality of the DNA (e.g., DNA from old samples or DNA mixed-up with normal cells) as it increases the background noise and makes detecting gene copy numbers more difficult [13]. As of today, NGS testing has been expanded to comprehensive genomic profiling (CGP), and to different tumor types, as *MET* amplification is a rare oncogenic driver across all solid tumor types. The only pitfall is that *MET* polysomy and *MET* amplification may not be distinguished by some NGS assays and do not control for CEP7. Therefore, if possible *MET* amplification is detected, it preferably needs to be confirmed by in situ hybridization (ISH) [40].

For instance, in NSCLC, there is a wide diversity of alterations leading to *MET* exon 14 skipping; therefore, enrichment for NGS is necessary. Amplicon or hybrid-capture-based DNA NGS was initially used so as not to miss low-frequency alleles in a broad area of interest around *MET* exon 14, increasing the detection rate by up to 2.6%. Additionally, RNA testing increased the positive testing ratio to 3.9%. The reason RNA testing resulted in higher positive samples was that it only needed to detect exon 13–15 fusion mRNA creating the MET-driven phenotype of NSCLC [41]. Most of this experience was based on exon-14-skipping mutations and cannot be generalized to gastric cancer.

### 3.2. Clinical Exposure of cMET-Driven Therapies in Gastric Cancer

#### 3.2.1. Monoclonal Antibodies

Rilotumumab is a monoclonal antibody that binds to the hepatocyte growth factor (HGF) and prevents it binding to the cMET receptor. In 2014, a randomized phase 2 study with rilotumumab was investigated in combination with epirubicin-cisplatin and capecitabine (ECX). Rilotumumab improved PFS in the combined arm with ECX to a median of 5.7 months compared to 4.2 months compared to placebo. The objective response rate (ORR) was 39% and the disease control rate (DCR) was 80% in the combined rilotumumab group [42]. Because of a distinct difference in effect between patients with high MET expression compared to patients with low MET expression, a subsequent phase 3 trial was set up: RILOMET-1 and RILOMET-2 (in Asia). In 2017, a first-line phase 3 study with rilotumumab was investigated (RILOMET-1) in standard of care therapy in patients overexpressing MET. Patients were screened with MET immunohistochemy; ≥25% of tumor cells with membrane staining of ≥1+ staining intensity were eligible. Different MET-detection tools were analyzed, e.g., the previously mentioned H-score, *MET* amplification via the MET/CEP ratio of 2 or more, as well as the average of MET copies < or >5. A total of 1477 patients were screened, of which 1043 (81%) were deemed MET positive. A total of 608 patients were randomly assigned to receive rilotumumab plus ECX or placebo plus ECX [43].

The study protocol was stopped early because of a higher number of deaths in the rilotumumab group. In an additional analysis, no biomarker (MET IHC, amplification, MET copies) could show a distinctive effect of the investigational drug. Preliminary analysis showed that rilotumumab was ineffective, with a median OS of 9.6 months compared to 11 months in the chemotherapy-alone group. Because of the high number of deaths, RILOMET II was closed shortly afterwards [43].

Another first-line phase 2 randomized clinical trial was performed in 2014 in the MEGA trial comparing standard of care chemotherapy folfox to folfox + panitumumab or folfox + rilotumumab. No pre-set diagnostic cMET-assay was included, although cMET staining and *MET* amplification were reported within all patients. IHC was positive in 60% of all patients, though no further details were given about the threshold for MET positivity. A total of 162 patients were included in the study. *MET* amplification was detected in 3 out of a total of 100 patients. Progression-free survival and overall survival was comparable between all arms; so, no added benefit was shown of rilotumumab or panitumumab in the first-line treatment of gastro-esophageal cancer [44].

In the METGastric trial, onartuzumab was evaluated in a randomized phase 3 study in a first-line combination with a backbone of mFOLFOX6. Onartuzumab is a recombinant, fully humanized, monovalent monoclonal antibody that binds with the extracellular domain of cMET. It prevents HGF from binding to the cMET-receptor and, therefore, restricts intracellular signaling. In an earlier phase II trial, the combination onartuzumab-erlotinib (EGFR inhibitor) resulted in an improved overall survival in patients with non-small-cell lung cancer (NSCLC) who were cMET positive, defined as 50% of tumor cells staining with an IHC intensity of 2+/3+. Therefore, the same screening with immunohistochemistry (IHC) and a 50% cell ratio was used in this phase 3 trial in gastric cancer. The proportion of patients with higher expression intensity was almost doubled in this study (38% vs. 21%) compared to the RILOMET-1 study. Unfortunately, the phase 3 part of the study was terminated early as the parallel phase 2 part could not ascertain the right patient selection. From the patients eligible for analysis, no difference in ORR could be found between standard of care and addition of onartuzumab [45].

A phase 2 non-randomized single-arm trial included 65 patients with advanced gastric cancer treated with emibetuzumab, an immunoglobulin G4 monoclonal bivalent anti-cMET antibody that blocks cMET signaling by blocking ligand-dependent cMET activation, as well as internalizing the cMET-receptor to be degraded in a ligand-independent manner. Patients were included beyond progression in second-line chemotherapy and screened for MET protein expression by immunohistochemistry (IHC) to be 2+ or 3+ positive in more than 60% of tumor cells. Of the 15 patients that were included, no patients had a partial response, apart from one patient with a −22% reduction in the target lesion, although he developed ascites and was considered progressive disease. *MET* amplification by ISH, defined by a MET/CEP7 ratio of ≥2, was found in 3 out of 4 patients with high IHC expression, although no relation could be found between any diagnostic marker and outcome [46].

#### 3.2.2. Tyrosine Kinase Inhibitors

A phase 1b study with AMG337 confirmed an overall response ratio (ORR) of 29.7% in *MET*-amplified patients with acceptable toxicity. AMG337 is a highly selective and potent small molecule inhibitor of cMET receptor signaling. A subsequent phase 2 study in patients with advanced esophagogastric cancer and other solid tumors who had received prior therapy was set up. Screening for *MET* amplification was performed by a central laboratory defined as a MET/CEP-7 ratio > 2.0. Over 2000 patients were screened of which 132 (6%) had a *MET* amplification; finally, 55 patients with measurable disease were enrolled. An ORR of 19% was reached in the cohort of esophago-gastric cancer, though not in other tumor types, but the study was terminated early as the study product could not uphold the earlier seen ORR of up to 62% in a small cohort of 13 patients. Among all the patients included in the analysis, the mean MET/CEP7 ratio was 7.7 (2.4–12.0) in the 8 responders and 7.1 (2.0–20.4) in the 39 non-responders; therefore, biomarker analysis did not show an association between the level of *MET* gene amplification and response to treatment. Possibly, the full potential of AMG337 could not have been investigated as this was a single-arm study and early termination likely influenced the final evaluation [47].

Crizotinib is a small molecule oral inhibitor of the anaplastic lymphoma kinase (ALK), c-MET/hepatocyte growth factor receptor (HGFR), and ROS receptor tyrosine kinases. Crizotinib is approved for *ALK* and *ROS1* gene rearrangement in non-small-cell lung cancer (NSCLC). Several cases with *MET* amplification in esophagogastric cancer showed efficacy for crizotinib for which a phase II trial was designed. MET overexpression was determined by central testing and initial screening was based on an IHC of 2+ or 3+; *MET* amplification was assessed by FISH, and the number of cMET gene copies for inclusion was set at ≥6 copies. cMET was prospectively analyzed in 570 patients with esophageal/junction or gastric adenocarcinoma and *MET* amplification was found in 35 patients (=6.1%). Finally, 11 patients were enrolled, of which 9 patients started therapy with crizotinib. The median copy number of MET was 7 (range 6–11). A tumor response rate was achieved in 2/3 patients (67%) with a low tumor MET amplification, and in 3/6 (50%) with an intermediate tumor MET amplification. Unfortunately, the trial was prematurely stopped due to insufficient accrual [48].

Capmatinib is an oral-type Ib cMET inhibitor that was studied within solid tumors of which nine patients with gastric cancer were treated. Patients were eligible after third-line treatment and inclusion was allowed based on immunohistochemistry (IHC), H-score > 150, MET/CEP7 ratio of ≥2 or a gene copy number of ≥5. Only two out of nine patients reached stable disease and no clear correlation between the different diagnostic techniques of MET overexpression and response was observed [49].

Foretinib, is a small-molecule multikinase inhibitor that targets MET, RON, AXL, TIE-2 and VEGFR2 receptors. It binds in the adenosine triphosphate pocket of its targets, resulting in conformational change and kinase inhibition. Foretinib has been evaluated in a phase 2 single-arm multicentric study with two dose cohorts (intermittent vs. continuous dosing). No diagnostic cMET biomarker was mandatory to enter the study. Patients eligible for the study had progressed beyond first-line chemotherapy. Only three patients had *MET* amplification and an additional 22% increased copy number due to polysomy. Across both arms, no patients experienced a partial response (PR) and 15 patients had stable disease (SD). There was no difference in response rate between patients with *MET* amplification and/or polysomy compared to patients without a cMET-driven biomarker. The median PFS was 1.7 months, while the median OS was 7.4 months with intermittent dosing and 4.3 months with daily dosing. This suggested that cMET signaling may not be critical in patients with gastric cancer without *MET* amplification [50].

Tivantinib is a low-molecular-weight, orally available selective inhibitor of cMET. It disrupts cMET phosphorylation in a non-ATP competitive manner. It was studied in an open-label phase 2 single-arm multicenter trial. A total of 30 patients were included without the necessity of a pre-diagnostic cMET assay. Pretreatment of one or two systemic therapies was allowed. A total of 11 patients achieved disease control, although only stable disease was reached. The median PFS was 43 days, and the median survival time was 344 days. As for the earlier studies, although IHC and FISH was performed, no biomarker was correlated with clinical benefit [51].

In 2017, a phase 1 run-in trial in solid tumors, and later phase 2, specifically for first-line esophago-gastric cancer, evaluated tivantinib in combination with folfox. A total of 34 patients was included in the phase 2 part and no pre-study diagnostic biomarker was necessary. The overall response ratio was similar to historic cohorts of standard chemotherapy folfox and no additional effect could be attributed to the addition of tivantinib. Moreover, no relation could be seen between IHC or MET protein expression [52].

The VIKTORY UMBRELLA trial is a basket study of patients with gastric cancer based on clinical sequencing and focuses on eight different biomarker groups of which *MET* amplification and MET overexpression (IHC 3+) are two. In this study savolitinib, a class I cMET inhibitor and small molecule receptor tyrosine kinase inhibitor was used. The study investigated the targeted therapy in second-line and compared it to second-line paclitaxel/ramucirumab. The incidence of MET overexpression by IHC (3+) was 8.8% (42/479) in this group, while 17 (40.5%) of 42 MET-overexpressed tumors had *MET*-amplified tumors by next-generation sequencing (NGS) or FISH, and 25 (59.5%) patients had no *MET* amplification. The overall response ratio in the *MET*-amplified arm was 50% (10 of 20). Patients with high MET copy number (≥10 MET gene copies by tissue NGS) had high response rates to savolitinib. One patient with upfront peritoneal metastases could be curatively resected after downstaging with savolitinib and was still alive one year after surgery. This shows that truly *MET*-amplified gastric cancer patients can gain an additional benefit from targeted therapy and even show better overall survival compared to patients receiving standard of care second-line chemotherapy [53].

Other TKIs have been studied in different tumor types but no extensive data in gastric cancer can be found. Cabozantinib has been investigated thoroughly in renal cell carcinoma (RCC) and hepatocellular carcinoma (HCC). It is a blocker of VEGFR-1 to 3, and the TAM family (TYRO3, AXL, MER), as well as cMET [54]. In the diffuse type gastric cancer cell line, one study investigated cabozantinib and showed strong targeting of cMET and VEGFR-2, suggesting its pivotal role. A study in all-comers 3d-line gastroesophageal cancer is now open, combining pembrolizumab-cabozantinib (NCT04164979), as well as the CAMILLA trial evaluating cabozantinib/durvalumab, with or without tremelimumab (NCT03539822).

Tepotinib is a selective cMET inhibitor that interrupts the cMET signal transduction pathway. In the VISION trial in NSCLC, tepotinib 500 mg in 152 patients showed a partial response in 50% of patients with the *MET* exon-14-skipping mutation resulting in an ORR of 46%. EMA recently approved the use of tepotinib in *MET* exon-14-skipping non-small-cell lung cancer [55].

One small interventional trial was performed in a cohort of solid tumors refractory to standard therapy. No MET diagnostic assay was necessary to enter the study. IHC and FISH analysis were determined throughout the study. Two patients with gastric cancer entered the study and one patient reached a PFS of 4, 6 months into therapy after four prior lines of chemotherapy. Therefore, the efficacy of tepotinib in gastric cancer is still unknown [56]. An overview of all clinical studies in gastric cancer with cMET inhibitors is given in Table 1.

## 4. Discussion

Among all the studies performed within cMET in gastric cancer, not many successes have been achieved. The RILOMET-1, MEGA and METGastric study all included patients in first-line gastroesophageal cancer and did not show a significant benefit in addition to chemotherapy.

One of the possible reasons why rilotumumab and onartuzumab did not show an increased benefit in first line is explained by the mechanism of action. It is a ligand-dependent antibody and in *MET* amplification the cMET pathway is autonomously active regardless of the effect on the binding ligand.

Secondly, screening by positive immunohistochemistry for cMET is not an appropriate biomarker. Overexpression based on immunohistochemistry did not show the good correlation with cMET-activity as an oncogenic driver typically seen in *MET* amplification. Moreover, in gastric cancer *MET* amplifications are still limited to around 5% of all patients, and, therefore, using cMET-targeted therapy for all first-line patients with gastric cancer results in overtreatment in patients without a *MET* amplification. This also means that the efficacy of these cMET-specific treatments could potentially be underrated [57,58,59].

In the study with AMG337, a potential benefit in patients with gastric cancer with an ORR of 19% could be shown. Patients were included based on a FISH MET/CEP7 ratio ≥ 2.0, but no distinct difference was found in the MET/CEP7 ratio in the responder and non-responder groups. In the study with crizotinib and capmatinib, screening was also based on IHC, and prospective analysis of *MET* gene amplification by GCN ≥ 5 or the MET/CEP7 ratio ≥ 2 could not distinguish responders from non-responders. However, this study had a low number of patients, and as mentioned earlier, the gene copy number is possibly not a good marker to detect *MET* amplification.

Foretinib was studied without cMET-specific inclusion criteria based on a diagnostic assay but was monitored in the study. Three patients had a *MET* amplification, but, across all patients, no partial responses were seen. A total of 15 patients showed stable disease. Again, a low number of patients was included in the study to determine a clinically significant benefit. Tivantinb was combined with a folfox regimen in a phase 1, and later in a phase 2, open-label trial. Although, in total, 49 patients were treated, no difference was seen compared to a historical cohort of folfox alone. No specific biomarker analysis could distinguish patients in terms of outcome.

The VIKTORY umbrella trial showed the most successful response for savolitinib on cMET-inhibitors for patients with gastric cancer and compared this to paclitaxel/ramucirumab second-line therapy. *MET* amplification was determined by NGS or FISH and an ORR of 50% was reached, showing the possible potential for cMET inhibitors in gastric cancer in a well-defined population.

## 5. Conclusions

All the forementioned studies had a different inclusion strategy, e.g., *MET* overexpression, rarely, a FISH MET/CEP7 ratio ≥ 2, or no prescreening assay was needed. Studies without an inclusion strategy could not find an overt biomarker predicting cMET sensitivity. Diagnostic assays of *MET* overexpression/amplification are not as homogeneous in gastric cancer compared to non-small-cell lung cancer (NSCLC), where the *MET* exon-14-skipping mutation has been extensively investigated and a clear strategy has been set out.

No beneficial effect has been seen in addition to chemotherapy for allcomers in first-line gastric cancer. In the cohorts with TKIs, all patient groups remain limited, and the inclusion strategies all remain different, apart from the clear analysis of the VIKTORY umbrella trial, which detected most patients based on NGS analysis with a good ORR of 50%.

The most important problem remains the need for clinically meaningful cut-off points, including the level of *MET* amplification as well as *MET* overexpression, to determine treatment-related decision-making.

Determining *MET* amplification by NGS or whole-exome sequencing could be the most accurate technique to predict *MET*-inhibitor sensitivity; however, confirmation by ISH will be important to distinguish true *MET* amplification from polysomy, depending on the type of NGS assay. More data on comprehensive genomic testing and *MET* inhibitors will be needed in other (basket) studies and further investigation will be needed to identify optimal predictive biomarkers under targeted therapy.

## Figures and Tables

**Figure 1 cancers-15-01976-f001:**
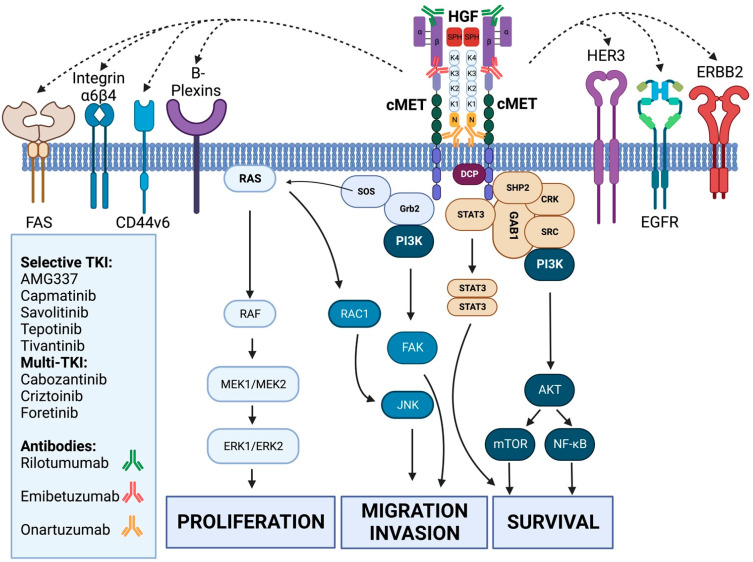
An overview of all the cMET inhibitor compounds with corresponding activity to the cMET pathway. HGF (hepatocyte growth factor) binds to cMET (mesenchymal-epithelial transition factor) with low- and high-affinity binding sites that induces cMET homodimerization and autophosphorylation by canonical activation. This activates GAB1, SRC, CRK, SHP2 and STAT3, where, together with PI3K, this leads to activation of AKT, the mTOR and NF-kB signaling cascades that control cell survival. Via Grb2 and SOS, the RAS/RAF and MAPK pathway is activated, inducing cell proliferation. Grb2 also activates PI3K, which, via FAK, controls migration/invasion. RAS by itself activates RAC1 that activates JNK, also responsible for cell migration and invasion. cMET can functionally be influenced by ERBB2, HER3 and EGFR receptors and launch synergistic activation. cMET is also regulated by different cell remodelers, such as FAS, integrin alpha 6 beta 4, CD44v6 and B-plexins, that facilitate the activation of the cMET tyrosine kinase domains and downstream cascade. (Created with BioRender.com, accessed on 21 November 2022).

**Table 1 cancers-15-01976-t001:** Overview of all studies that have been performed in gastric cancer with MET-inhibitors.

Type	Target	Name	Mechanism of Action	Trial (Ref.)	Phase	N° of Patients Included	Inclusion Diagnostic Marker	Effect
Monoclonal antibody	HGF	mFolfox + Rilotumumab	Blocks HGF	Rilomet I [19]	III	608	IHC ≥ 1 ≤	No benefit of mAb
		mFolfox + Rilotumumab or EGFRi		MEGA [21]	III	162	IHC 2+ or 3+	No benefit of mAb
	MET	Onartuzumab + mFolfox	Blocks MET	METGastric [22]	III	562	none	No benefit of mAb
	HGF and MET	Emibetuzumab	Blocks HGF binding and internalization of MET	NA [23]	II	15	IHC 2+ or 3+	No patients with PR
Tyrosine kinase inhibitor		Crizotinib	Multi TKI	AcSé Crizotinib Program [25]	II	9	IHC 2+ or 3+	ORR 33%
		Foretinib		NA [27]	II	74	None	No PR, 15 patiënts SD
		Capmatinib	Selective MET TKI	NA [26]	II	9	IHC 2+ or 3+ or Hscore > 150 MET/CEN7 ≥ 2 or MET GCN ≥ 5	2 patients with SD
		Tivantinib		NA [28]	II	30	None	No ORR, 36 DCR
		Savolitinib		Viktory Umbrella Trial [30]	II	20	Amplification ≥ 10 on NGS	ORR 50%
		AMG337		NA [24]	II	55	MET/CEN7 ≥ 2	ORR 19%

ORR: objective response ratio; SD: stable disease; PR: partial response; mAb: monoclonal antibody; TKI: tyrosine kinase inhibitor; DCR: disease control rate; IHC: immunohistochemistry; GCN: gene copy number; NGS: next-generation sequencing.

## Data Availability

The data can be shared up on request.

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
