# Peer review of "The Role of cMET in Gastric Cancer—A Review of the Literature"

_cancers, 2023, doi:10.3390/cancers15071976_

Round 1

Reviewer 1 Report

The review ‘The role of cMet in gastric cancer….’ by Filip Van Herpe and Eric Van Cutsem summarizes the role of cMet in cancer and issues with its detection in clinical setting and the outcomes from Clinical Trials targeting cMet or cMet associated signaling pathways.

Comments:

1)      Please add a Figure or scheme depicting the role of cMET-mechanism of action/effect on activation of signaling pathways in gastric cancer or other cancers in general.  Additionally, please do add a sub-section (write-up) on the mechanism-as of now only a few lines have been dedicated to it.

2)      It will add value to the review if a small subsection on the in vitro or pre-clinical studies is also written; as of now it is very difficult to comprehend the importance and the outcomes of pre-clinical studies that have led to clinical trials targeting cMET.

3)      Describe abbreviated names as they appear in the text…for example what is S1 (line 45).

4)      Since a majority of the review focus is on difficulty encountered during IHC etc., and the staining intensity-it will add value to the review to add staining images. The reviewer understands that that it needs access to copyright material-but it would be worth the effort!

5)      When introducing a new therapeutic drug please explain the class or role/target of that drug (especially in studies where combinations trials have been written) so that the reader can have a better understanding of why the combination was selected.

6)      In the write-up on Clinical trials, please explicitly write which cancer type the trial was focused on; this is because, in certain sections the authors refer to other cancers (such as lung cancer) and  it is not clear which cancer type are the authors referring to.

7)      There are issues with sentence formation (correct placement of commas etc) and other grammatical areas; please address these issues as these may lead to misinterpretation of scientific details.

8)      Certain section the authors refer to cMET while in certain sections it is referred to as MET only; please clarify if there is a justification for this or correct as relevant.

Author Response

1) Please add a Figure or scheme depicting the role of cMET-mechanism of action/effect on activation of signaling pathways in gastric cancer or other cancers in general.  Additionally, please do add a sub-section (write-up) on the mechanism-as of now only a few lines have been dedicated to it.

I updated the figure and added some extra notes of the cMET pathway and cMET mechanism.

2)      It will add value to the review if a small subsection on the in vitro or pre-clinical studies is also written; as of now it is very difficult to comprehend the importance and the outcomes of pre-clinical studies that have led to clinical trials targeting cMET.

A small subsection of preclinical studies have been added with a better build up to why cMET was used as a target in gastric cancer.

3)      Describe abbreviated names as they appear in the text…for example what is S1 (line 45).

S1 is a correct abbreviation of S-1 as chemotherapy (5FU analogue in Asia for gastric cancer).

4)      Since a majority of the review focus is on difficulty encountered during IHC etc., and the staining intensity-it will add value to the review to add staining images. The reviewer understands that that it needs access to copyright material-but it would be worth the effort!

Unfortunately as MET overexpression is not standard of care molecular testing in gastric cancer so this information is scarce and copyright material will take a long time to get this sorted.

5)      When introducing a new therapeutic drug please explain the class or role/target of that drug (especially in studies where combinations trials have been written) so that the reader can have a better understanding of why the combination was selected.

6)      In the write-up on Clinical trials, please explicitly write which cancer type the trial was focused on; this is because, in certain sections the authors refer to other cancers (such as lung cancer) and  it is not clear which cancer type are the authors referring to.

This was changed in materials and methods.

7)      There are issues with sentence formation (correct placement of commas etc) and other grammatical areas; please address these issues as these may lead to misinterpretation of scientific details.

I changed the sentence formation throughout the whole manuscript. Hopefully this makes the interpretation better.

8)      Certain section the authors refer to cMET while in certain sections it is referred to as MET only; please clarify if there is a justification for this or correct as relevant.

This was changed in to cMET in all sections. 

Reviewer 2 Report

In the study, the authors reviewed The role of cMET in gastric cancer given the importance of cMET. however, the cMET review was not innovative and the number of citations in the article was not sufficient to qualify as a review. The authors need to continue to add to the data and papers and provide a more innovative perspective on cMET as well.

The author is recommended to make a thorough revision. Several concerns need to be addressed by the authors, as summarized below.

Major concerns:

1. What are the mutation status, gene expression, and prognosis of cMET in gastric cancer?

2. Please show the role of cMET in gastric cancer, including pathways and functions, using a schematic diagram.

3. Add more extensive literature coverage to support the full-text arguments.

Author Response

  1. What are the mutation status, gene expression, and prognosis of cMET in gastric cancer?

A more updated report on gene expression and prognosis for cMET in gastric cancer has been provided

  1. Please show the role of cMET in gastric cancer, including pathways and functions, using a schematic diagram.

An updated figure of cMET in gastric cancer has been provided with according pathways.

  1. Add more extensive literature coverage to support the full-text arguments.

More data has been added on the preclinical work and pathway analysis although for clinical results on MET directed therapies, no additional data is available.

Reviewer 3 Report

This is a nice review. However it is a bit too focused and could have been broader in scope by including perhaps some other biomarkers. Consequently, the number of bibliographic entries is a bit low for a review.

Minor revision

Introduction

Line 36,37 and 42: there is a redundancy. Same concept in very close sentences

Line 47: what GEC stand for? It is GC (gastric cancer)?

LINE 54: “overall survival (OS)” could be simple changed to “OS “since the acronymous was introduced two lines before (line 52)

Results

The description of cMET in the Results section 3.1 (Lines from 76 to 88) would sound more appropriate in the introduction.

Line 126-7: change post-transcription to post-transcriptionally

Lines from 110 to 127 should be a single paragraph without a line break.

Line 131: “Capuzzo criteria”, bibliography is missing.

Line 144: 3.1.1. is 3.1.2

Line 159: cut “only”

Results paragraph 3.2.1.: at the end of the paragraph a brief summary of the published results with monoclonal antibody would be of utility. (like: in summary….)

Line 227: “E/GEJ/G “the acronymous should be written in full the first time is used.

Line 278: “GEJ” the acronymous should be written in full the first time is used

Author Response

Dear colleague,

I do see that the number of references might be low for a review. based on additional analyses by other reviewers extra reviews will be added.

Introduction

Line 36,37 and 42: there is a redundancy. Same concept in very close sentences.
I removed line 36-37 as this was redundant.

Line 47: what GEC stand for? It is GC (gastric cancer)? 
changed it to GC (gastric cancer)

LINE 54: “overall survival (OS)” could be simple changed to “OS “since the acronymous was introduced two lines before (line 52). 
This was changed

Results

The description of cMET in the Results section 3.1 (Lines from 76 to 88) would sound more appropriate in the introduction.

I deliberately did not add the whole description in introduction as this could be too extensive, but changed  the first lines in 76 to the introduction.

Line 126-7: change post-transcription to post-transcriptionally

Changed this

Lines from 110 to 127 should be a single paragraph without a line break.

Was changed

Line 131: “Capuzzo criteria”, bibliography is missing.

ref = added

Line 144: 3.1.1. is 3.1.2

Changed

Line 159: cut “only”

changed

Results paragraph 3.2.1.: at the end of the paragraph a brief summary of the published results with monoclonal antibody would be of utility. (like: in summary….)

Personally i believe that the table 1 is showing a clear overview of all studies and it's best outcomes.

Line 227: “E/GEJ/G “the acronymous should be written in full the first time is used.

was changed

Line 278: “GEJ” the acronymous should be written in full the first time is used

was changed

Reviewer 4 Report

Authors presented a review on the role of cCMET proto-oncogene in gastric cancer (GC). They discuss the principles of treatment of GC in the advanced stage in a comprehensive but at the same time sufficient way. The principles of determining cMET overexpression and its target role in the treatment of GC patients are clearly presented. Studies with the use of cMET inhibitors are discussed thoroughly, explaining in detail the mechanism of action of individual drugs.

The need for further research and the extension of the cMET expression test with the NGS method in order to select patients for further research was clarified. 

Author Response

Dear colleague,

Thank you for reading this review on cMET in gastric cancer. No additional comments are necessary.

Round 2

Reviewer 3 Report

The work has been sufficiently improved to be accepted for publication.

Author Response

Dear Reviewer,

Thank you for the positive feedback